# Exposure to SARS-CoV-2 generates T-cell memory in the absence of a detectable viral infection

Zhongfang Wang[1,6], Xiaoyun Yang [1,6], Jiaying Zhong[1,6], Yumin Zhou[1,6], Zhiqiang Tang[2,6], Haibo Zhou[3], Jun He[4], Xinyue Mei [1], Yonghong Tang[4], Bijia Lin[1], Zhenjun Chen [5], James McCluskey [5], Ji Yang[1], Alexandra J. Corbett [5] & Pixin Ran [1✉]

T-cell immunity is important for recovery from COVID-19 and provides heightened immunity for re-infection. However, little is known about the SARS-CoV-2-specific T-cell immunity in virus-exposed individuals. Here we report virus-specific CD4[+] and CD8[+] T-cell memory in recovered COVID-19 patients and close contacts. We also demonstrate the size and quality of the memory T-cell pool of COVID-19 patients are larger and better than those of close contacts. However, the proliferation capacity, size and quality of T-cell responses in close contacts are readily distinguishable from healthy donors, suggesting close contacts are able to gain T-cell immunity against SARS-CoV-2 despite lacking a detectable infection. Additionally, asymptomatic and symptomatic COVID-19 patients contain similar levels of SARS-CoV-2-specific T-cell memory. Overall, this study demonstrates the versatility and potential of memory T cells from COVID-19 patients and close contacts, which may be important for host protection.

[1] State Key Laboratory of Respiratory Disease & National Clinical Research Center for Respiratory Disease, Guangzhou Institute of Respiratory Health, the First Affiliated Hospital of Guangzhou Medical University, Guangzhou Medical University, Guangzhou, China. [2] The Second Peoples Hospital of Changde City, Hunan, China. [3] The Sixth Affiliated Hospital of Guangzhou Medical University, Guangzhou, China. [4] Affiliated Nanhua Hospital of University of South China, Hunan, China. [5] The Department of Microbiology and Immunology and The Peter Doherty Institute for Infection and Immunity, University of Melbourne, Melbourne, Victoria, Australia. [6] These authors contributed equally: Zhongfang Wang, Xiaoyun Yang, Jiaying Zhong, Yumin Zhou, Zhiqiang Tang. ✉email: pxran@gzhmu.edu.cn

Since early 2020, SARS-CoV-2 has spread globally, triggering a pandemic that continues to cause devastating damage to public health and people's livelihoods. By the middle of November, the global COVID-19 cases have reached 50 million with the death toll exceeding a grim 1.2 million (John Hopkins University, USA). Although the mechanisms by which host immunity combats SARS-CoV-2 infection are far from being completely understood, significant knowledge in this area has been gained through the investigations of the association of COVID-19 clinical features and disease progression with host immune responses[1]. For example, our recent study established that the severity of COVID-19 inversely correlates with T-cell immunity of the host[2]. In the presence of adequate neutralizing antibodies, CD4+ and CD8+ T cells play a major role in the recovery of critical COVID-19 patients[2]. Other studies showed that in moderate and severe COVID-19 cases characterized by lymphopenia there was a drastic reduction in the numbers of both CD4+ and CD8+ T cells[3–5]. Although the reason for this reduction remains unknown, autopsy revealed extensive infiltration of T cells into the lungs[6]. Analysis of immune cells from bronchoalveolar lavage (BAL) fluid of COVID-19 patients demonstrated the presence of clonal expansion[7]. Moreover, virus-specific CD4+ T cell numbers were shown to be associated with the production of IgG that targets the receptor-binding domain (RBD) of SARS-CoV-2[8]. Notably, analyses of persistent COVID-19 cases showed that upon activation their T-cells appeared to lose polyfunctionality and cytotoxicity, trending towards an exhausted phenotype[9,10].

While most acute viral infections result in the development of protective immunity, available data suggest that long-term and robust-protective memory is not easily acquired for human coronavirus infections[11]. For example, one year after disease onset following MERS-CoV infection, the viral-specific IgG antibody became undetectable for some of the patients with mild symptoms[11–13]. The SARS-CoV-1 humoral response was relatively short-lived and memory B cells disappeared quickly after primary infection[14]. Recent mathematical modeling suggested a short duration (likely <2 years) of protective immunity is elicited after SARS-CoV-2 infection[15]. Furthermore, Long et al. have reported that the viral-specific IgG levels of SARS-CoV-2-infected individuals had an ~70% reduction during the early convalescent phase and a significant proportion of individuals (40% of asymptomatic patients and 12.9% of symptomatic patients) became IgG seronegative[16]. In contrast to the short-lived humoral response in SARS-CoV-1 survivors, the magnitude and frequency of specific CD8+ memory T cells, and to a lesser extent CD4+ memory T cells, persisted for 6–11 years, suggesting that T cells may confer long-term immunity[15]. Although it has been reported that SARS-CoV-2-specific CD4+ and CD8+ T cells were detected in 100 and 70% of convalescent COVID-19 patients, respectively[17], to date, it remains largely unclear how well the SARS-CoV-2 T cell memory is established and how the memory T cells respond upon re-exposure to viral antigens. Another important question that remains unresolved is whether close contacts, who had been confirmed to be negative in nucleic acid testing (NAT) and antibody screening, have gained any memory T cell immunity upon exposure to SARS-CoV-2.

In this study, we examined the proliferation and activation capability of the SARS-CoV-2 memory T cell pools of a large cohort of recovered COVID-19 patients, close contacts, and unexposed healthy individuals. Our results showed that the COVID-19 patients and close contacts developed SARS-CoV-2-specific T-cell immune memory. In addition, comparable levels of SARS-CoV-2-specific memory T cells were detected in the samples of asymptomatic and symptomatic COVID-19 patients.

## Results

**Proliferation capacity of memory T cells from recovered COVID-19 patients and close contacts.** To assess the SARS-CoV-2-specific T-cell memory, human peripheral blood mononuclear cells (PBMCs) from 90 COVID-19 patients collected between 48–86 days after disease onset were stimulated in vitro for 10 days with peptide pools designed to target the spike glycoprotein (S), membrane glycoprotein (M), nucleocapsid (N), envelope glycoprotein (E) and ORF1ab region of RNA-dependent RNA polymerase (RdRp) of SARS-CoV-2. Our data showed that the memory CD4+ and CD8+ T cells of 94.44% and 83.33%, respectively, of the COVID-19 patients successfully underwent expansion (Fig. 1a–c). These results clearly indicate that most of the recovered COVID-19 patients have developed effective T cell memory pools against SARS-CoV-2.

Although the close contacts in our cohort were all negative in both nucleic acid test (NAT) and SARS-CoV-2 antibody screening, the possible exposure of these individuals to the virus may have led to the generation of T cell immunity even in the absence of a successful infection. To test this possibility, we performed a 10-day in vitro peptide stimulation assay for 69 close contacts from 45 family clusters. The results show that 57.97% (Fig. 1a–c) and 14.49% (Fig. 1b, c) of close contacts contained virus-specific memory CD4+ and CD8+ T-cells, respectively. Notably, all close contacts developed responses at lower frequencies than 4%, while 64 (71.11%) and 32 (35.56%) of the 90 COVID-19 patients developed marked responses at the frequencies of higher than 4% for IFNγ+CD4+ T cells (Fig. 1a) and IFNγ+CD8+ T cells (Fig. 1b), respectively. In comparison to the COVID-19 patients, a significantly lower proportion of close contacts responded ($p < 0.0001$ for CD4+, Fig. 1a; $p < 0.0001$ for CD8+, Fig. 1b).

In order to investigate whether the observed expanded T cells may have originated from pre-existing cross-reactive T cells specific for common cold coronaviruses from previous infections, we tested blood samples of 63 healthy donors collected before September of 2019. Following a 10-day in vitro peptide expansion only 3.17% of the healthy donors contained detectable levels of virus-specific memory CD4+ and CD8+ T cells, respectively (Fig. 1a–c), suggesting that cross-reactive T cells derived from exposure to other human coronaviruses do exist but are at a significantly lower frequency than those observed in close contacts.

The major differences between the proportion of COVID-19 patients and healthy donors ($p < 0.0001$ for CD4+, Fig. 1a; $p < 0.0001$ for CD8+, Fig. 1b), or between close contacts and healthy donors ($p < 0.0001$ for CD4+, Fig. 1a; $p = 0.0157$ for CD8+, Fig. 1b) with memory T-cells capable of proliferating in response to SARS-CoV2 peptides emphasize that exposure to SARS-CoV-2 can facilitate the establishment of the T memory immunity not only in COVID-19 patients, but also in some close contacts even in the absence of a successful infection. In addition, differences between COVID-19 patients and close contacts were observed in the frequency of double-positive (IFNγ+ TNF+) CD4+ T cells ($p < 0.0001$ for CD4+, Supplementary Fig. 1a, $p < 0.0001$ for CD8+, Supplementary Fig. 1b), although CD4+, but not CD8+ cells producing both cytokines were significantly higher in close contacts than healthy controls (Supplementary Fig. 1a, b).

**Ex vivo analyses of SARS-CoV-2-specific memory T cells from COVID-19 patients and close contacts.** Next, we measured the sizes of virus-specific memory pools for CD4+ and CD8+ T cells from 89 COVID-19 patients (1 COVID-19 sample was used up), 69 close contacts and 30 healthy donors by using an overnight "ex vivo" peptide stimulation assay. Our results demonstrated that a significant proportion of COVID-19 patients contained

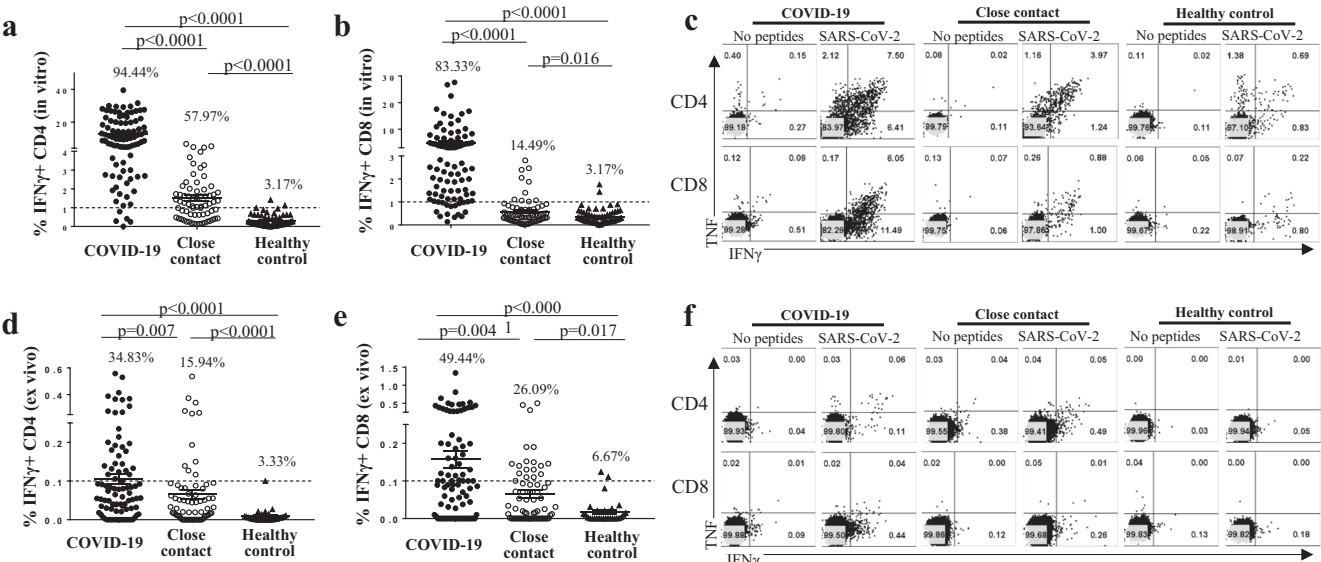

**Fig. 1 Memory T cells specific to SARS-2 were detected and can proliferate in vitro in COVID-19 patients and in close contacts.** Donor PBMCs were stimulated with 15-mer peptide pools (overlapping by 11 amino acids) encompassing the entire spike (S), nucleocapsid (N), membrane (M), and envelope (E) proteins for 10 days (in vitro expansion, **a–c**) or overnight (ex vivo, **d–f**) in the presence of 10 U/ml rIL-2, IFNγ, and TNF expressing cells were enumerated by intracellular cytokine staining. Ninety COVID-19 patients (closed circle), their 69 close contacts (open circle), and 63 unexposed healthy donors (closed triangle) were assayed in vitro. For ex vivo experiments, the samples from the above cohort except for one from the COVID-19 group because of cells used up, and 30 of the 63 unexposed healthy donors were assayed. Graphs show the frequency of IFNγ expressing cells in (**a**) CD4+ and (**b**) CD8+ T cells after in vitro expansion and overnight stimulation and in (**d**) CD4+ and (**e**) CD8+ T cells after overnight stimulation. Dashed line is the cut off determined by the background staining (no peptide) for the healthy control group. The cut off threshold used for the overnight stimulation experiments was based on all negative controls (95% CI). The percentages shown are the frequency above this cut off. **c, f** Representative dot plots showing IFNγ and TNF expression in T cells after expansion (**c**) or overnight stimulation (**f**). **a, b, d, e** Error bars indicate mean frequencies of IFNγ+ T cells ± SEM; Percentage shown on top of the plots indicates the frequencies of samples above the cutoff. The student t test was performed with two-sided p values as indicated. No peptides: no peptide stimulation control. SARS-CoV-2: with stimulation by SARS-CoV-2 overlapping peptide pools.

virus-specific T cells (34.83% for CD4+, Fig. 1d; 49.44% for CD8+, Fig. 1e; and cut off = 0.1%) at 48–86 days after disease onset. In addition, SARS-CoV-2-specific T cells were also detected in close contacts (15.94% for CD4+, Fig. 1d and 26.09% for CD8+, Fig. 1e). Significant differences were seen between the sizes of T cell memory pools of COVID-19 patients and close contacts (p = 0.007 for CD4+, Fig. 1d and p = 0.004 for CD8+, Fig. 1e). In contrast, in the case of the healthy donors, we found that only 1/30 (3.33%) and 2/30 (6.67%) of the samples contained cross-reactive memory CD4+ and CD8+ T cells, respectively (Fig. 1d, e), suggesting that the cross-reactive T-cell immunity only exists in a small number of unexposed healthy donors. Interestingly, comparing the frequency of double-positive (IFNγ+ TNF+) CD4+ and CD8+ T cells within individuals, these were higher in both COVID-19 patients and close contacts than in healthy controls (Supplementary Fig. 1c, d).

**IFNγ-producing SARS-CoV-2-specific memory T cells are detectable in close contacts of infected individuals.** To evaluate the quality of SARS-CoV-2-specific memory T cells, we measured the MFI of IFNγ by intracellular staining in the memory T cells from COVID-19 patients and close contacts. To increase the robustness of this experiment, we included an internal control where all of the samples were also assessed for the production of IFNγ following stimulation with CMV peptide pools spanning the pp65 protein. From the comparison between the MFI values of the different samples, it is clear that; (i) CMV peptides induced similar levels of IFNγ production by CD4+ and CD8+ T cells in the samples from COVID-19 patients and close contacts (Fig. 2a, c, e), (ii) the expression levels of IFNγ in CMV-specific T cells were 2-3 times higher than those of SARS-CoV-2-specific CD4+ (Fig. 2c)

or CD8+ T cells (Fig. 2e); (iii) SARS-CoV-2 peptides induced higher levels of IFNγ production in both CD4+ (Fig. 2b, c) and CD8+ (Fig. 2d, e) T cells from patients infected with COVID-19 compared with close contacts, the MFIs being twice as high in CD4+ T cells from the infected group. Collectively, these results indicate that the activation capability of SARS-CoV-2-specific memory T cells from close contacts is lower than that in the COVID-19 patients, despite both groups having similar pre-existing immunity to CMV.

**Memory T-cell immunity is detectable in both symptomatic and asymptomatic patients with COVID-19 infection.** PBMCs from 72 symptomatic and 18 asymptomatic COVID-19 patients were used in the overnight ex vivo and 10-day in vitro expansion assays to evaluate the sizes, qualities and proliferation capacities of the memory T cell pools. Data in Fig. 3a, d show that following overnight stimulation by peptide pools, 4/18 (22.22%) and 7/18 (38.89%) of the samples from the asymptomatic patients with COVID-19 developed detectable numbers of SARS-CoV-2 specific IFNγ-producing CD4+ T cells and CD8+ T cells, respectively. For the symptomatic COVID-19 patients, 27/71 (35.23%) and 36/71 (50.70%) of the samples also developed virus-specific specific CD4+ T cells and CD8+ T cells, respectively (Fig. 3a, d). There was no significant difference in the sizes of the SARS-CoV-2-specific memory T-cell pools between the symptomatic and asymptomatic COVID-19 patients (p = 0.58 for CD4+ and p = 0.66 for CD8+, Fig. 3a, d). Meanwhile, the ex vivo analysis showed that the MFI of IFNγ staining of the memory T cells (SARS-CoV-2-specific) from the asymptomatic and symptomatic patients were 1536.37 ± 165.28 and 1182.18 ± 219.92 for CD4+ (Fig. 3b) and 636.54 ± 56.25 and 578.47 ± 102.37 for CD8+

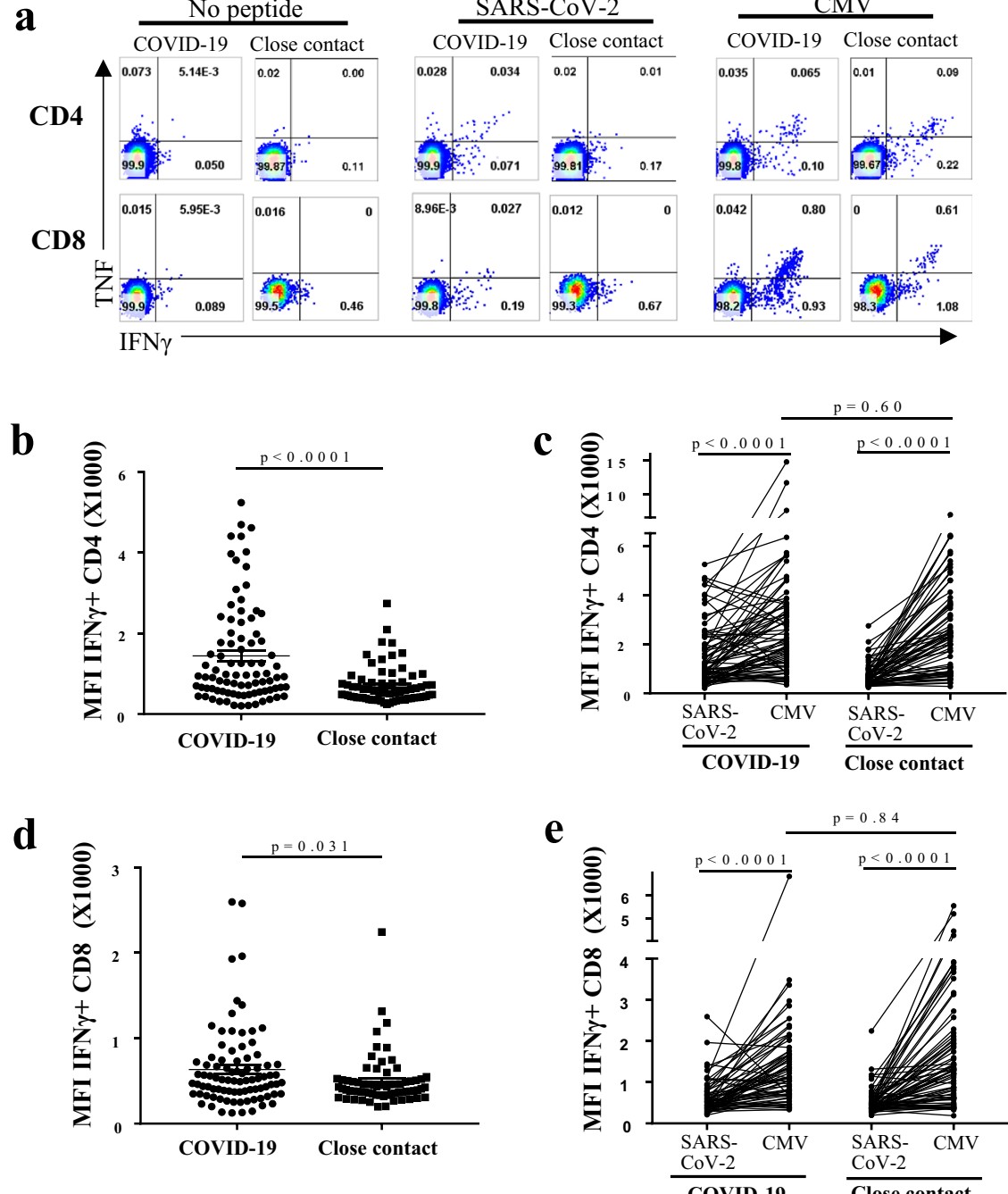

**Fig. 2 Functional analysis of SARS-CoV-2 specific memory T cells in Covid-19 patients and close contacts.** Donor PBMCs were stimulated with SARS-CoV-2 or CMV 15-mer peptide pools overnight in the presence of 10 U/ml rIL-2, IFNγ and TNF expressing cells were enumerated by intracellular cytokine staining. **a** Representative FACS plots showing the expression of IFNγ and TNF in CD4+ and CD8+ T cells with or without SARS-CoV-2 or CMV peptide stimulation overnight, as indicated. **b, d** Mean Fluorescence Intensity (MFI) of IFNγ staining for (**b**) CD4+and (**d**) CD8+ T cells from COVID-19 patients (close circle, $n = 89$) and their close contacts (closed square, $n = 69$) after overnight stimulation with SARS-CoV-2 peptide pool. **c, e** Paired analyses of MFI for IFNγ of CD4+ (**c**) and CD8+ (**e**) T cells after overnight stimulation with SARS-CoV-2 or CMV peptide pools for COVID-19 patients ($n = 79$) and close contacts ($n = 69$). Each symbol represents a data point from one individual. **b–e** Error bars represent mean ± SEM. The student $t$ test was performed with two-sided $p$ values as indicated. No peptides: no peptide stimulation control. SARS-CoV-2: with stimulation by SARS-CoV-2 overlapping peptide pools. CMV: with stimulation by CMV overlapping peptide pools.

(Fig. 3e), respectively. Thus, there was no significant difference in the qualities of the memory T cells between the asymptomatic and symptomatic patients ($p = 0.39$ for CD4+ and $p = 0.44$ for CD8+, Fig. 3b, e).

In vitro peptide stimulation and expansion showed that 88.89% and 72.22% of CD8+ T cells from the symptomatic and asymptomatic patients, respectively, proliferated to detectable levels (Fig. 3f). For the CD4+ T cells, 97.22% and 83.33% of the samples from the symptomatic and asymptomatic patients, respectively, proliferated to levels above 1% (Fig. 3c). This indicates a slightly reduced proliferation capacity in SARS-CoV-2-specific T-cell immunity of asymptomatic patients ($p < 0.0001$, Fig. 3c).

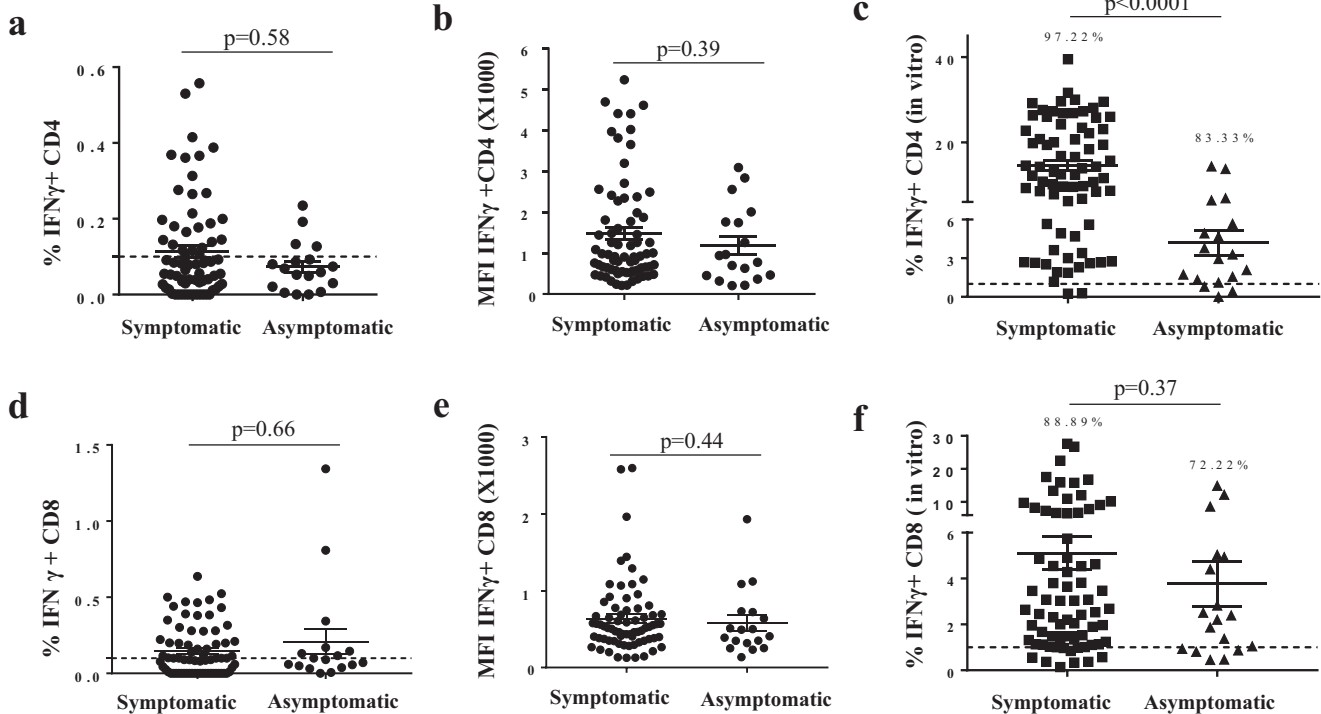

**Fig. 3 Comparisons of the T cell memory and in vitro expansion of SARS-CoV-2 specific T cells between symptomatic and asymptomatic COVID-19 patients. a**, **d** Frequencies of IFNγ expressing cells in CD4$^+$ (**a**) and CD8$^+$ (**d**) T cells in recovered symptomatic ($n = 71$) and asymptomatic ($n = 18$) COVID-19 patients ex vivo. **b**, **e** MFI of IFNγ staining in CD4$^+$ (**b**) and CD8$^+$ (**e**) T cells from symptomatic and asymptomatic COVID-19 patients. **c**, **f** Frequencies of IFNγ expressing (**c**) CD4$^+$ and (**f**) CD8$^+$ T cells after in vitro expansion in symptomatic ($n = 72$) and asymptomatic ($n = 18$) COVID-19 patients. Percentages shown are the frequencies above the cut off (1%, which was the upper limit observed in no-peptide control stimulations). Error bars represent mean ± SEM. The student's $t$ test was performed with two-sided $p$ values indicated.

**SARS-CoV-2-specific T cells are stably maintained 48–86 days after onset of symptoms**. We then examined if there was any correlation between the magnitude of the T cell responses (measured by an in vitro expansion assay) and the timespan between 48 and 86 days after symptom onset and found no relationship between the levels of SARS-CoV-2-specific T cells (CD4$^+$ and CD8$^+$) and the timespan within this period ($R^2 = 0.025$, $p = 0.14$ for CD4$^+$, Supplementary Fig. 2a, and $R^2 = 0.005$, $p = 0.52$ for CD8$^+$, Supplementary Fig. 2c). Meanwhile, our data also showed that there was no association between the levels of memory T cells measured by an ex vivo assay and the timespan between 48–86 days after disease onset ($R^2 = 0.064$, $p = 0.021$ for CD4$^+$, Supplementary Fig. 2b and $R^2 = 0.066$, $p = 0.019$ for CD8$^+$, Supplementary Fig. 2d). Together, our in vitro and ex vivo data suggest that CD4$^+$ T memory and CD8$^+$ T memory may have contracted to a stable plateau by the times these samples were collected. Furthermore, we also did not see any difference between severe COVID-19 and moderate COVID-19 patients in the proportion of SARS-Co-V2-specific IFNγ-producing CD4$^+$ or CD8$^+$ T cells expanded in vitro ($p = 0.71$ for CD4$^+$, Supplementary Fig. 2e, $p = 0.48$ for CD8$^+$, Supplementary Fig. 2f).

**Memory CD4$^+$ T-cell responses correlate with IgG titers against N protein and S RBD of SARS-CoV-2**. The neutralizing antibody response in MERS-CoV-2 infection was previously shown to be dependent on the CD4$^+$ T cell response[13]. To determine if this is also true for SARS-CoV infection, we performed correlation analyses between IgG titers (anti N and anti-RBD, Supplementary Table 1) and magnitude of memory T cells measured by in vitro and ex vivo assays. The sensitivity and accuracy of assays for IgG measurements were verified as shown

in Supplementary Table 2. Following in vitro expansion the virus-specific memory CD4$^+$ T cell pool correlated with the titers of IgG against the S RBD region ($R^2 = 0.51$, $p < 0.0001$, Supplementary Fig. 3a) and the N protein ($R^2 = 0.48$, $p < 0.0001$, Supplementary Fig. 3b), whereas no apparent correlation between CD8$^+$ T cells and IgG titers was observed ($R^2 = 0.28$, $p < 0.0001$, anti-S RBD IgG, Supplementary Fig. 3c and $R^2 = 0.28$, $p < 0.0001$, anti-N IgG, Supplementary Fig. 3d). In the ex vivo assay, no correlation was found between either the virus-specific CD4$^+$ T cells and IgG titres ($R^2 = 0.01$, $p = 0.27$ anti-S RBD IgG, Supplementary Fig. 3e and $R^2 = 0.01$, $p = 0.29$, anti-N IgG, Supplementary Fig. 3f) or the virus-specific CD8$^+$ T cells and IgG titres ($R^2 = 0.03$, $p = 0.10$, anti-S RBD, Supplementary Fig. 3g and $R^2 = 0.03$, $p = 0.10$, anti-N IgG, Supplementary Fig. 3h), indicating that, due to the low numbers of specific T cells that can be detected ex vivo in the memory phase, expansion of T cells in vitro to increase their numbers may be necessary to observe these correlations.

**Discussion**
COVID-19 patients display a wide range of clinical phenotypes, including severe, moderate, mild, and asymptomatic cases, likely determined by a mix of host genetic factors, and the dose and route of infection. Individuals also exhibit a wide variation in cellular and humoral immune responses during the primary viral infection, with some patients displaying balanced viral-specific B cell and T cell immunity, whereas others rely either on a higher level of activation of neutralizing antibodies or on a stronger T cell response to fight off the virus[2]. In rare cases, individuals who suffer severe and long-lasting symptoms show highly imbalanced

cellular and humoral immune responses whereby the levels of SARS-CoV-2 specific T-cell or antibody immunity are very low[2].

Close contacts, who are SARS-CoV-2-exposed, are often both NAT negative and antibody negative, indicating that SARS-CoV-2 failed to establish a successful infection within these individuals, presumably due to their exposure to limited numbers of viral particles or a short time of exposure. However, our analysis of the samples from 69 of these close contacts showed the presence of SARS-CoV-2 specific memory T-cell immunity. A similar observation was reported during the MERS epidemic where high-risk individuals (e.g., camel workers) who were NAT negative and antibody negative also developed significant levels of MERS-CoV specific memory T cells[13]. In addition, although in agreement with Sekine et al.[18], we found that some polyfunctional T cells were detectable in close contacts, cells producing both IFNγ and TNF appear largely specific for infected patients rather than for close contacts and healthy donors, suggesting that for COVID-19 patients, the occurrence of stronger antigen stimulation and greater inflammation during viral infection led to an enhanced polyfunctional T-cell response.

Our ex vivo stimulation analyses demonstrated that the pool sizes and quality of the SARS-CoV-2-specific CD4+ and CD8+ T memory cells from close contacts were around half of those from COVID-19 patients. Similarly, our in vitro expansion experiments showed that the SARS-CoV-2-specific CD4+ memory T cells of 57.97% and 94.44% of close contacts and COVID-19 patients, respectively, were able to proliferate. However, a more remarkable difference between the CD8+ proliferation frequencies of the two sample groups was observed, such that the SARS-CoV-2-specific CD8+ memory T cells of 14.49% and 83.33% of close contacts and COVID-19 patients, respectively, underwent proliferation. Theoretically, the initial activation of SARS-CoV-2-specific CD8+ and formation of CD8+ T memory are achieved through the endogenous pathway which processes viral antigens produced within the virus-infected host cells[19]. Presumably, without in situ replication of SARS-CoV-2, there are insufficient viral antigens within the host cells of close contacts to induce a robust CD8+ response resulting in CD8+ T memory in the majority of individuals. By contrast, the formation of CD4+ T memory does not rely on endogenous viral replication but involves endocytosis and/or phagocytosis of exogenous viral antigens, which are mostly derived from non-replicative viral particles or soluble viral proteins[19]. Thus, CD4+ T cell memory may be more easily achieved in uninfected exposed individuals.

Initially, we observed that SARS-CoV-2-specific memory CD4+ and CD8+ secreted low levels of IFNγ and only a small proportion of the T cells from COVID-19 patients gained multifunctionality (IFNγ and TNF dual expression). To vigorously validate this finding, we analysed the CMV-specific memory T cells in the same PBMC samples. Evidently, the levels of IFNγ and TNF expression and the numbers of CMV-specific CD4+ and CD8+ memory T cells were all significantly greater than those of the corresponding SARS-CoV-2-specific memory T cells (Fig. 2a), ruling out the possibility that SARS-CoV-2 infection inhibits the function of T cells of the host. Recent epidemiological data show that between 18 and 62% of SARS-CoV-2 infections are asymptomatic[20–23]. Therefore, determining how well protective immunity is established in asymptomatic COVID-19 patients will provide valuable information for understanding herd immunity and the design of strategies to combat secondary infections by the virus. To this end, we compared the T-memory immunity levels between asymptomatic and symptomatic COVID-19 patients and showed that the sizes and quality of their memory pools are comparable. Only the in vitro expansion capacity of memory CD4+ from asymptomatic COVID-19 patients was significantly lower. Since our data showed the magnitude of in vitro expansion of CD4+ memory T cells is

correlated to the IgG titers of anti-RBD and anti-N, it is possible that the antibody production of asymptomatic individuals is lower than that of symptomatic individuals. This observation is consistent with the findings that there is a rapid decay of anti-SARS-CoV-2 antibodies and IgG antibodies in asymptomatic patients[24].

In agreement with recent reports[17,25], our data also demonstrated the presence of cross-reactive memory CD4+ and CD8+ T cells, which target various surface proteins of SARS-CoV-2, in unexposed healthy donors. However, the failure of these cross-reactive memory CD4+ and CD8+ to expand in vitro suggests they have limited potential to function as part of a protective immune response against SARS-CoV-2. It is noteworthy that the SARS-CoV-2-reactive T cells detected in the unexposed healthy donors in our study were lower than those detected by Grifoni et al.[17] and Braun et al.[26], but were consistent with those reported by Peng et al.[27] and Zhou et al.[28]. Assumably, due to the use of different methodologies in assessing SARS-CoV-2-specific T-cell responses, it is difficult to directly reconcile the cell-number data between different studies. Thus, a thorough investigation is needed to determine whether the cross-reactive T memory can provide any protective immunity and exert an influence on the outcomes of COVID-19 disease.

In summary, by examining a substantial number of clinical samples, we determined the SARS-CoV-2-specific memory T-cell immunity in COVID-19 patients with various clinical symptoms. Despite some subtle differences, most patients developed measurable amounts of SARS-CoV-2-specific CD4+ and CD8+ memory T cells which were stably maintained between 48–86 days after convalescence. Importantly, our discovery of the presence of significant levels of SARS-CoV-2-specific memory T-cell immunity in a group of individuals (close contacts) who were exposed to but not infected by the virus highlights some unique characteristics in the dynamic interactions between SARS-CoV-2 and its human host. Although cross-reactive memory T cells were present in healthy donors who had never been exposed to SARS-CoV-2, their role in host protection needs to be thoroughly investigated as they were hardly able to proliferate. Together, our analyses add important information on the landscape of immune responses of a range of individuals in response to the primary SARS-CoV-2 encounter during the first wave of the pandemic.

## Methods

**COVID-19 patients, close contacts, and healthy donors**. For this study, we recruited 90 COVID-19 patients and 69 close contacts. All of the COVID-19 patients (NAT+) had stayed in the hospital and then recovered. The medical data collected from the COVID-19 patients included symptoms at disease onset and records of physical examinations, laboratory tests and imaging. Asymptomatic COVID-19 patients were defined using strict criteria: they were negative for any signs of cough, fever, sore throat, runny nose or computed tomography (CT) image changes in the lungs. A blood sample was taken from each of the patients in the period between d48 and d86 after disease onset or returning a NAT+ result.

Close contacts were identified from family members or friends who had stayed with a SARS-CoV-2 infected individual(s) at the time from 5 days before their disease onset to hospitalization. They were classified as a close contact only if they also were within a close distance (<1.5 m) of a COVID-19 individual(s) in a confined space for >1 h or were living together with a known case for >24 h. Other important criteria were that they were NAT- and negative for SARS-CoV-2-specific antibodies (IgG and IgM) against S RBD and/or N and virus neutralization tests. For this study, a blood sample was taken from each of the close contacts at the time d48 and d86 after exposure to a known COVID-19+ individual.

Blood samples of 63 healthy donors were obtained from a local blood donation center in September 2019 (before the start of the COVID-19 pandemic) for unrelated studies. These donors were considered healthy as they had no known history of any significant systemic diseases. As the blood samples from healthy donors were frozen for a longer period of time compared to those from patients and close contacts, we assessed whether prolonged freezing had any effect on assay outcomes by comparing the CMV-specific T-cell responses (which would be expected to be the same) of close contacts and healthy donors (HC) in a control experiment. We found that there is no significant difference in the frequencies of CMV-specific CD4+ and CD8+ between the two groups of samples (CD4+: $p = 0.32$ and CD8+: $p = 0.37$).

This study is approved by the Ethics Commission of the First Affiliated Hospital of Guangzhou Medical University (No.2020-51). The signed consent forms from all the participants were obtained.

**Peptide pool design and preparation.** SARS-CoV-2-specific peptides were designed and synthesized as follows. The protein sequences were derived from the SARS-CoV-2 reference (GenBank: MN908947.3). Four hundred and forty-seven 15-mer SARS-CoV-2 epitopes (overlapping by 11 amino acids) spanning the entire antigen region of spike (S), nucleocapsid (N), membrane (M), and envelope (E) proteins were generated with an online peptide generator (Peptide 2.0), and were synthesized by GL Biochem Corporation (Shanghai) with a purity of over 80%. One hundred and ten 18-mer peptides (overlapping by 10 amino acids) encompassing the ORF1ab region of RNA-dependent RNA polymerase (RdRP) were synthesized by GL Biochem Corporation (Shanghai). Each peptide was dissolved in DMSO, and was then pooled, with each at a concentration of 45 μM to form a stock.

**PBMC isolation and ex vivo stimulation.** PBMCs were isolated from heparinized whole blood by density-gradient sedimentation using Ficoll-Paque according to the manufacturer's instructions (GE Healthcare, 17-1440-02). $1 \times 10^6$ PBMCs were cultured in RPMI 1640 medium (Gibco) supplemented with 10% heat-inactivated FBS (Biological Industries, Israel Beit-Haemek), 100 U/ml penicillin (Gibco) and 0.1 mg/ml streptomycin (Gibco). The PBMCs were treated with the peptide pool containing 447 15-mer peptides and 110 18-mer peptides at 125 nM/each peptide in the presence of 10 U/ml rIL-2 and 1 μM GolgiPlug (BD Biosciences, San Diego, CA) overnight at 37 °C, 5% $CO_2$. The approach of using a large peptide pool to stimulate PBMCs was based on that developed by Chevalier M. F. et al.[29] and was validated for CMV peptides.

**PBMC in vitro expansion culture and stimulation.** For in vitro culturing and stimulation, $1 \times 10^6$ PBMCs were treated with the peptide pool (125 nM/each peptide), and incubated for 10 days. During this culturing, half of the medium was changed twice per week with fresh PRMI 1640 supplemented with 10% FBS and 10 U/ml rIL-2. The cells were subcultured when needed. The cells were then re-stimulated at day 10 with a medium containing the peptide pool (125 nM/each peptide) overnight before being stained for FACS analysis.

**Flow cytometry.** Cells harvested from the overnight or 10-day stimulation cultures were washed and incubated with Live/dead aqua V510 for 15 min on ice. Cells were then washed again and surface-stained for 30 min on ice with the following antibodies: anti-CD3-FITC (BioLegend, clone UCHT1, 1:200, Cat# 300406), anti-CD4-APC-Cy7 (BD Pharmingen™, clone RPA-T4, 1:200, Cat# 561839), anti-CD8-PerCPCy5.5 (BD Bioscience, clone RPA-T8, 1:200, Cat# 560662). After fixation and permeabilization with Cytofix and Perm (BD Bioscience, Cat# 554714) on ice for 15 min, intracellular staining (ICS) was performed on ice for 30 min with anti-TNF-PE-Cy7 (BD, clone MAb11, 1:200, Cat # 557647) and anti-IFNγ-APC (BD Pharmingen™, clone B27, 1:200, Cat# 554702). After the final wash, cells were resuspended in 200 μl FACS buffer. The samples were acquired using an FACSAria III instrument (BD Bioscience) and analyzed with FlowJo software (Treestar).

**Detection of blood plasma IgG in COVID-19 patients and close contacts.** The SARS-CoV-2-specific IgG in the blood plasma was detected with two ELISA kits targeting N protein and S protein RBD, separately (Guangzhou Darui, China), and one chemiluminescent immunoassay kit targeting N plus S protein (Shenzhen YHLO Biotech, China). The IgG levels specific to N plus S protein was also determined by using a lateral flow immunochromatographic assay kit (DIAGREAT, Beijing, China). For immunochromatographic assays, the optical signal was quantified with a time-resolved immunochromatographic analyzer and was calculated according to established programmed standards. The cut off value for the assignment of positive samples was determined according to the manufacture's instructions. An individual was considered seropositive if a positive result was generated by all three assays.

**Reporting summary.** Further information on research design is available in the Nature Research Reporting Summary linked to this article.

## Data availability

All relevant data are available from the authors.

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

## Acknowledgements

This work was supported by the National Key Basic Research Project (2019YFC0810900), Ministry of Science and Technology of P.R. China, and NSFC 81971485, Guangdong Key Basic Research Project 2019B1515120068, Guangdong Key Research and Development Project 2020B1111330001.

## Author contributions

Z.W. and P.R. designed the experiments, analysed data, and wrote the paper; X.Y., J.Z., and X.M. performed the experiments and analysed data; Y.Z., Z.T., H.Z., J.H., Y.T., and B.L. recruited the cohort and carried out clinical treatments; Z.C., J.C., J.Y., and A.C. reviewed this work and wrote the paper.

## Competing interests

The authors declare no competing interests.
