## [Peer Review File · Nature Communications]

REVIEWER COMMENTS

Reviewer #1 (Remarks to the Author):

An interesting and provocative paper that suggests that T cell immune responses may be induced in close contacts of SARS-CoV-2 infected patients.
The discussion is balanced and important in relation to the potential role of cross-reactivity.
The work is well performed technically.

Questions

- the samples from the healthy donors would have been frozen from 2019. The samples from patients and close contacts appear to have been analysed as fresh samples. Have the team assessed how freezing can influence assay outcomes ?
- incorporation of this large number of peptides in one stimulation is unusual. To what extent might it underestimate responses due to peptide competition ?
- the negative cut off on the overnight stimulation (Figure 1D,E) seems high and may underestimate responses.
- the combination of IFN-TNF double staining appears largely specific for infected patients and some discussion as to the physiological basis for this would be useful.
- the recent paper by Sekine should be discussed.

Reviewer #2 (Remarks to the Author):

There is great urgency to careful reports of findings of Ab versus T cell immunity in different settings of COVID-19 infection. This manuscript offers a dataset of relevance to this debate, considering symptomatic convalescent, mild, asymptomatic and control groups at the level of Ab response and CD4, CD8 responses by ICS to large peptide pools.

After the first few papers, the debate has necessarily become more nuanced around longevity of response, evidence of correlates of protection, cross-reactivity, and the thorny question of 'how much is enough?'

This manuscript has relevant data in these regards, yet I often found it over- or misinterpreted, with some really quite sweeping statements on big, controversial issues, not well backed by the data.

The biggest conundrum to resolve is that current papers attempt to sample the T cell repertoire using different approaches that give different (and not always quantitatively comparable) answers, including ICS and ELISpot. This paper uses a rather different protocol to any of the others, culturing large cell numbers (a million cells/well) for 10 days in a pool >500 peptides and supplemented with 10IU rIL-2, then measuring CD4 or CD8 cytokine shift on ICS. The responder cells are referred to as 'memory' but so far as I could see, no memory subset markers are included in the panels. The measured responses are clearly specific, disease-related and are a biomarker of response, though it is impossible to relate the absolute meaning of the cell numbers here to other papers, due to the 10 day expansion in exogenous IL-2. There are some quirks to this assay system, since the frequency of responders to in pre-COVID PBMC are way lower than I have seen in any other study.

For reassurance, it would be good to see some tethering of this assay system to more conventional techniques: do they get the same answers in the different cohorts if they leave out the 10d+IL-2 pre-culture and simply do a conventional assay by ELISpot or ICS? This seems really important, as the system seems so amplified for signal in this study.

There were many quirks and inaccuracies in the writing which made the message hard to get at.

The notion of a response being common between individuals was frequently conflated with it being high-frequency within individuals (though, as mentioned, this is hard to debate when cells have been expanded for 10 days in vitro).

There were some rather rash and unsupported statements in the Abstract and Discussion. I could not see how the data gave a 'hint for population immunity and vaccine strategy' - this was not justified at any point.

The paragraph from line 286 about x-reactive cells being present but unable to expand made little sense as a counter-argument to any role in protection. The proposed caveat about high HLA diversity of HLA genotypes being a special confounder was a rabbit-hole that to me offered no sense.

Point by Point Response:

Reviewer 1:

1. *The samples from the healthy donors would have been frozen from 2019. The samples from patients and close contacts appear to have been analysed as fresh samples. Have the team assessed how freezing can influence assay outcomes?*

All samples were frozen before assays were carried out. Indeed, the blood samples from healthy donors were frozen for a longer period of time compared to those from patients and close contacts and so the reviewer raises an important point. Yes, we did assess whether prolonged freezing had any effect on assay outcomes by comparing the CMV-specific T-cell responses of close contacts and healthy donors (HC) in a control experiment, since these would be expected to be equivalent between these groups. We found that there is no significant difference in the frequencies of CMV-specific CD4⁺ and CD8⁺ between the two groups of samples (CD4⁺: $p=0.32$ and CD8⁺: $p=0.37$), indicating that prolonged freezing should not have influenced assay outcomes. We have added this information into the Materials and Methods (lines 330 and 335).

2. *Incorporation of this large number of peptides in one stimulation is unusual. To what extent might it underestimate responses due to peptide competition?*

A conventional protocol usually involves the stimulation of PBMCs ($\sim 5 \times 10^5$) with up to 50 peptides at the concentration of 2 μM (for each peptide) in multiple wells. However, in this study, due to the large cohort sizes (90 patients, 69 close contacts and 60 healthy donors), the presence of relatively small numbers of PBMCs in each sample and our use of two stimulation experiments, it was impractical for us to use the above mentioned method for the analysis of our samples. Thus, we modified this method so that we could compare the total virus-specific CD4⁺ or CD8⁺ response in one well for stimulation with peptides covering main target proteins. A

CMV peptide pool was used to optimize the stimulation protocol. We first divided the 383 peptides covering main T cell targeting (PP65, IE1 and IE2) into 8 pools with the concentration of 2 μ M for each peptide. The sum of 8 pools of stimulated CMV-specific CD4⁺ or CD8⁺ T cells was set as the standard for subsequent comparison. We then combined the 383 peptides into one pool with each peptide added at varying concentrations (62.5 nM, 125 nM, 250 nM and 500 nM) and performed a stimulation assay. We found that at concentration of 125 nM, the 383-peptide mix yielded comparable levels of virus-specific CD4⁺ and CD8⁺ T-cell responses to those of the sum of 8 peptide pools at the concentration of 2 μ M. Thus, 125 nM/each peptide was used throughout our analyses of SARS-CoV-2-mediated T-cell responses with 447 15-mer peptides and 110 18-mer peptides. Our careful calibration of the assay system should have allowed us to minimise any possible underestimation of the T cell responses. A similar approach has been developed to monitor human tumor-specific T-cell responses with large peptide pools, where the optimal concentration of each peptide in the 300-peptide pool for stimulation was 100 nM (very similar to the optimal concentration identified and used in our study) (Chevalier *et al.*, 2015). For clarification, we have cited this reference in the Materials and Methods (lines 358 and 359). We would be happy to show as supp data as asked or could leave as is just with the reference.

3. *The negative cut off on the overnight stimulation (Figure 1D, E) seems high and may underestimate responses.*

The cut-off threshold used for the overnight stimulation experiments (Figure 1D, E) was based on the background (no peptide) staining of all healthy controls (mean+3SD, 0.049 for CD4⁺ and 0.1 for CD8⁺) (lines 418-419). Although this cut off seems high, it is consistent for all samples and, therefore, will not affect the comparison of the responses between COVID-19 patients, close contacts and healthy donors.

It is hard to define the cutoff of effective T cell response for overnight stimulation experiments. In order to avoid overstatement, we define our cut-off threshold

4. *The combination of IFN-TNF double staining appears largely specific for infected patients and some discussion as to the physiological basis for this would be useful.*

We agree and have added the following paragraph into the Discussion (lines 271-276): “In addition, we found that, although some polyfunctional T cells were detectable in close contacts

in agreement with Sekine *et al.*, double staining of IFN γ and TNF α appears largely specific for infected patients rather than for close contacts and healthy donors, suggesting that in COVID-19 patients, the occurrence of a stronger antigen stimulation and greater inflammation during viral infection led to an enhanced poly-functional T-cell response.”

5. *The recent paper by Sekine should be discussed.*

We thank the Reviewer for reminding us of this highly relevant paper. We have now cited the Sekine paper and added the following sentence into the Discussion (lines 243-247): “Furthermore, a recent study by Sekine *et al.* showed that polyfunctional SARS-CoV-2-specific T cells were detectable in seronegative individuals who were exposed to the virus.”

Reviewer 2:

1. *After the first few papers, the debate has necessarily become more nuanced around longevity of response, evidence of correlates of protection, cross-reactivity, and the thorny question of 'how much is enough?' This manuscript has relevant data in these regards, yet I often found it over- or misinterpreted, with some really quite sweeping statements on big, controversial issues, not well backed by the data. The biggest conundrum to resolve is that current papers attempt to sample the T cell repertoire using different approaches that give different (and not always quantitatively comparable) answers, including ICS and ELISpot. This paper uses a rather different protocol to any of the others, culturing large cell numbers (a million cells/well) for 10 days in a pool >500 peptides and supplemented with 10IU rIL-2, then measuring CD4 or CD8 cytokine shift on ICS.*

Firstly, we wish to clarify that in this study we carried out two separate stimulation experiments, *in vitro* and *ex vivo* ICS assays. The *ex vivo* assay was performed using an overnight peptide stimulation, similar to previous studies. To make this clearer in the manuscript, we changed “by using a short-term “*ex vivo*” peptide stimulation assay” into “by using an overnight *ex vivo* peptide stimulation assay” in the *ex vivo* analyses section (line 135). In line 169, we changed “short-term *ex vivo*” into “the overnight *ex vivo*”. In the Materials and Methods, we added “*ex vivo*” in line 350 which now reads “PBMC isolation and *ex vivo* stimulation”. These changes

emphasize the overnight stimulation condition for the *ex vivo* assay, making it consistent with what has been stated in the Material and Methods and Figure Legend.

The 10-day *in vitro* stimulation was designed to enhance the sensitivity of the assay in order to detect any subtle differences between different cohorts and assess the proliferation capacity of SARS-CoV-2-specific T cells upon peptide stimulation. Although the *in vitro* and *ex vivo* assays yielded data with different baselines and scales, the differences in T-cell responses between the three cohorts are largely comparable (Fig. 1 A, B, D, E).

The question about the large peptide pool used in the assays is similar to that raised by Reviewer 1. This method is based on the one established by Chevalier M. F. *et al.* (OncoImmunology, 2015, 4:10, e1029702, High-throughput monitoring of human tumor-specific T-cell responses with large peptide pools). We have now cited this paper in the Materials and Methods (lines 358 and 359). In addition, before using the large peptide pool to stimulate SARS-CoV-2-specific T cells, we first calibrated our assay system and identified the optimal peptide concentration (125 nM) using CMV-specific peptides in a control experiment. Please also see response to Reviewer 1 question 2.

The same concentration of cells (1 million cells/well) has been used in other published studies for the analyses of T-cell responses upon infections by SARS-CoV-2 and other viruses (Peng *et al.*, 2020; Sekine *et al.*, 2020; Sridhar *et al.*, 2013) With the above clarification, we hope that we have addressed the Reviewer's concerns.

2. *The responder cells are referred to as 'memory' but so far as I could see, no memory subset markers are included in the panels.*

The blood samples used in this study were taken from convalescent COVID-19 patients and their close contacts between 48-86 days after the patients showed symptoms. According to data published by Peng *et al.* (Peng *et al.*, 2020), T cells detected by CD27 and CD45RA staining in convalescent COVID-19 patients 28 days after infection (20 to 60 days earlier than our samples) were mostly T memory cells. In response to the Reviewer's comment, we have applied CD27 and CD45RA staining to examine the representative samples used in our study and have shown that the SARS-CoV-2-specific T cells are predominantly central memory (T_{cm},

CD27⁺CD45RA⁻) and effector memory T cells (Tem, CD27⁻CD45RA⁻) (upper panel, and shown in red and overlaid with total CD4⁺ T cells in lower panel).

Phenotyping of virus-specific CD4⁺ T cells. 1X10⁶ PBMC cells were pulsed with SARS-CoV-2 peptide pool (125 nM). IFN γ ⁺ CD4⁺ T cells (blue in upper panel, red in lower panel) and CD4⁺ T cells (blue in lower panel) were stained for CD27 and CD45RA.

3. *The measured responses are clearly specific, disease-related and are a biomarker of response, though it is impossible to relate the absolute meaning of the cell numbers here to other papers, due to the 10 day expansion in exogenous IL-2. There are some quirks to this assay system, since the frequency of responders to in pre-COVID PBMC are way lower than I have seen in any other study. For reassurance, it would be good to see some tethering of this assay system to more conventional techniques: do they get the same answers in the different cohorts if they leave out the 10d+IL-2 pre-culture and simply do a conventional assay by ELISpot or ICS? This seems really important, as the system seems so amplified for signal in this study.*

We agree that, due to the use of different methodologies in assessing SARS-CoV-2-specific T-cell responses, it is difficult to directly reconcile the cell-number data between different studies. It is true that the SARS-CoV-2-reactive T cells detected in the un-exposed healthy control in our study were lower than those detected by Grifoni *et al.* (Grifoni *et al.*, 2020) and Braun *et al.* (Braun *et al.*, 2020), where an AIM (Activation-induced markers) assay was used. However, our results are consistent with those reported by Peng *et al.* (Peng *et al.*, 2020) and Zhou *et al.* (Zhou *et al.*, 2020) where, using the same IFN- γ based intracellular cytokine staining (ICS) assay, the IFN- γ -producing SARS-CoV-2-specific T-cell responses were not observed in 16 healthy

unexposed volunteers and spike-specific T-cell responses were not detected in 108 un-exposed volunteers. It is reasonable to assume that the differences are due to the different sensitivities of the AIM and ICS methods. IFN- γ ELISpot and ICS experiments pulsed with peptide pools are well-established methods for evaluating virus-specific T cells, mostly virus-specific CD8⁺ CTL and Th1 CD4⁺, which have been shown to correlate with protection or recovery (Sridhar *et al.*, 2013; Wang *et al.*, 2015). The AIM assay is based on surface activation markers independent of cytokine production and is capable of detecting early-responding T cells. However, which method can detect T cells that are more relevant to protection against second COVID-19 infection is currently unknown and requires a more detailed investigation.

We reiterate that the 10 day stimulation was not used for all assays – this appears to be a misunderstanding by the reviewer. Please see our full response to this in point 1, above). Briefly, Fig. 1 (A, B, D, E) showed the comparison of data from the 10-day stimulation with those of the overnight stimulation. As can be seen, although baselines and scales of the two sets of assay data are different, the relative differences in T-cell responses between the three cohorts are largely comparable.

4. *There were many quirks and inaccuracies in the writing which made the message hard to get at.*
 - a. *The notion of a response being common between individuals was frequently conflated with it being high-frequency within individuals (though, as mentioned, this is hard to debate when cells have been expanded for 10 days in vitro).*

We agree that the 10-day culturing cannot distinguish the common response between individuals from the high frequency part within individuals. Our 10-day culturing experiment was mainly designed to detect the proliferation capacity of SARS-CoV-2-specific T cells but the complementary *ex vivo* experiments (overnight stimulation and ICS) were readily able to detect the size of SARS-CoV-2 specific T cells, either from SARS-CoV-2 exposure-induced or cross-reactive T cells derived from human common coronavirus. As mentioned in our response to question 1 (Reviewer 2), we have made a number of changes in the text to clarify the overnight stimulation condition for the *ex vivo* assay, in accordance with the rationale that this experiment was used to measure the

sizes of virus-specific memory pools for CD4⁺ and CD8⁺ T cells from COVID-19 patients, close contacts and healthy donors (lines 135, 169 and 350).

b. There were some rather rash and unsupported statements in the Abstract and Discussion. I could not see how the data gave a 'hint for population immunity and vaccine strategy' - this was not justified at any point.

We agree and have removed these sentences from previous text: “providing a hint for population immunity and vaccine strategy” (line 41); “These findings have important implications with respect to future vaccination and population immunity” (lines 85-87).

c. The paragraph from line 286 about x-reactive cells being present but unable to expand made little sense as a counter-argument to any role in protection. The proposed caveat about high HLA diversity of HLA genotypes being a special confounder was a rabbit-hole that to me offered no sense.

To address this concern, we have deleted the following sentences from the paragraph: “In addition, given the high diversity of human HLA subtypes and the complexity of the immunodominance hierarchy, there is little chance for the common T-epitopes shared between SARS-CoV-2 and other human “common-cold” coronaviruses to become immunodominant. Even if they did, these epitopes would only be presented in a limited number of HLA subtype populations. As such,”

- Braun, J., Loyal, L., Frentsch, M., Wendisch, D., Georg, P., Kurth, F., . . . Thiel, A. (2020). SARS-CoV-2-reactive T cells in healthy donors and patients with COVID-19. *Nature*. doi:10.1038/s41586-020-2598-9
- Chevalier, M. F., Bobisse, S., Costa-Nunes, C., Cesson, V., Jichlinski, P., Speiser, D. E., . . . Derre, L. (2015). High-throughput monitoring of human tumor-specific T-cell responses with large peptide pools. *Oncoimmunology*, 4(10), e1029702. doi:10.1080/2162402X.2015.1029702
- Grifoni, A., Weiskopf, D., Ramirez, S. I., Mateus, J., Dan, J. M., Moderbacher, C. R., . . . Sette, A. (2020). Targets of T Cell Responses to SARS-CoV-2 Coronavirus in Humans with COVID-19 Disease and Unexposed Individuals. *Cell*, 181(7), 1489-1501 e1415. doi:10.1016/j.cell.2020.05.015
- Peng, Y., Mentzer, A. J., Liu, G., Yao, X., Yin, Z., Dong, D., . . . Dong, T. (2020). Broad and strong memory CD4(+) and CD8(+) T cells induced by SARS-CoV-2 in UK convalescent individuals following COVID-19. *Nat Immunol*, 21(11), 1336-1345. doi:10.1038/s41590-020-0782-6

- Sekine, T., Perez-Potti, A., Rivera-Ballesteros, O., Stralin, K., Gorin, J. B., Olsson, A., . . . Buggert, M. (2020). Robust T Cell Immunity in Convalescent Individuals with Asymptomatic or Mild COVID-19. *Cell*, *183*(1), 158-168 e114. doi:10.1016/j.cell.2020.08.017
- Sridhar, S., Begom, S., Bermingham, A., Hoschler, K., Adamson, W., Carman, W., . . . Lalvani, A. (2013). Cellular immune correlates of protection against symptomatic pandemic influenza. *Nat Med*, *19*(10), 1305-1312. doi:10.1038/nm.3350
- Wang, Z., Wan, Y., Qiu, C., Quinones-Parra, S., Zhu, Z., Loh, L., . . . Xu, J. (2015). Recovery from severe H7N9 disease is associated with diverse response mechanisms dominated by CD8(+) T cells. *Nat Commun*, *6*, 6833. doi:10.1038/ncomms7833
- Zhou, F., Yu, T., Du, R., Fan, G., Liu, Y., Liu, Z., . . . Cao, B. (2020). Clinical course and risk factors for mortality of adult inpatients with COVID-19 in Wuhan, China: a retrospective cohort study. *Lancet*, *395*(10229), 1054-1062. doi:10.1016/S0140-6736(20)30566-3

REVIEWERS' COMMENTS

Reviewer #1 (Remarks to the Author):

The authors have done a good rebuttal of my concerns.

Reviewer #2 (Remarks to the Author):

Overall, I find the manuscript considerably tidied and clarified. Ultimately, we're all trying to get our heads around the true meaning of the various sets of T cell data, and we're desperate for clarity in terms of who is making a meaningful response to what, and when. In this respect, I found some of the answers helpful (eg the memory characterisation) and some still somewhat argumentative (along the lines of 'others have reported extremely low frequencies from extremely high density cell cultures, so why not here..?')

REVIEWERS' COMMENTS

Reviewer #1 (Remarks to the Author):

The authors have done a good rebuttal of my concerns.

We thank the Reviewer for the comments and the time spent reviewing our manuscript.

Reviewer #2 (Remarks to the Author):

Overall, I find the manuscript considerably tidied and clarified. Ultimately, we're all trying to get our heads around the true meaning of the various sets of T cell data, and we're desperate for clarity in terms of who is making a meaningful response to what, and when. In this respect, I found some of the answers helpful (eg the memory characterisation) and some still somewhat argumentative (along the lines of 'others have reported extremely low frequencies from extremely high density cell cultures, so why not here..?')

We totally agree that it is important and urgent to understand the role of different T cell subset in anti- SARS-CoV-2 immunity. We believe, as more studies on T-cell responses to SARS-CoV-2 infection are completed, we will have a clearer picture of the various SARS-CoV-2-specific T cells, including their identity, when they function and what role they play.

We apologise for sounding argumentative in our response to the reviewer. This was not our intention. To address the reviewers' concern on this point we have now modified the concluding remarks in the last paragraph of the Discussion (lines 289-291). We toned the conclusion down from “Although cross-reactive memory T cells were present in healthy donors who had never been exposed to SARS-CoV-2, the fact that they were barely detectable and unable to proliferate makes them unlikely to be versatile for host protection.” To “Although cross-reactive memory T cells were present in healthy donors who had never been exposed to SARS-CoV-2, their role in host protection needs to be thoroughly investigated as they were hardly able to proliferate.” We also added some more discussion of other similar studies on page 11, line 275-281.

It is noteworthy that the SARS-CoV-2-reactive T cells detected in the un-exposed healthy control in our study were lower than those detected by Grifoni *et al.*²⁵ and Braun *et al.*²⁶, but

were consistent with those reported by Peng *et al.*²⁷ and Zhou *et al.*²⁸. Assumably, due to the use of different methodologies in assessing SARS-CoV-2-specific T-cell responses, it is difficult to directly reconcile the cell-number data between different studies. Thus, a thorough investigation is needed to determine whether the cross-reactive T memory can provide any protective immunity and exert an influence on the outcomes of COVID-19 disease.